# Diagnostic Approach to Equine Testicular Disorders

**DOI:** 10.3390/vetsci11060243

**Published:** 2024-05-29

**Authors:** Muhammad-Salman Waqas, Eduardo Arroyo, Ahmed Tibary

**Affiliations:** 1Comparative Theriogenology, Department of Veterinary Clinical Sciences, College of Veterinary Medicine, Center for Reproductive Biology, Washington State University, Pullman, WA 99163, USA; salman.waqas@wsu.edu; 2Department of Large Animal Clinical Sciences, College of Veterinary Medicine, University of Florida, Gainesville, FL 32610, USA; eduardoarroyo@ufl.edu

**Keywords:** stallions, testis, hernia, cryptorchidism, trauma, degeneration, scrotum, ultrasonography, biopsy, orchitis

## Abstract

**Simple Summary:**

Management of breeding stallions is crucial to equine reproduction. The long-life use of a stallion for a breeding career is the ultimate objective, whether it happens through natural mating or through semen collection and artificial insemination. Stud farm veterinarians should be aware of the techniques used to evaluate testicular function and the diagnostic approach to testicular disorders in cases of emergency. This paper presents the clinical methods used to assess testicular health, including palpation, ultrasonography, biopsy, and fine-needle aspiration. The discussion of testicular disorders is broken down into four categories: congenital (present at birth) disorders (cryptorchidism, monorchidism, and testicular hypoplasia), differential diagnosis of scrotal enlargement, differential diagnosis of causes of progressive testicular enlargement, and differential diagnosis of testicular asymmetry or reduction in size with an emphasis on testicular degeneration. Severe clinical signs often accompany a sudden increase in testicular size and are a major cause of stallions being referred for surgery. Testicular disorders are illustrated with clinical cases seen by the authors.

**Abstract:**

Management of breeding stallions is crucial to equine reproduction. The longevity of the breeding career is the ultimate objective, whether the stallion is used for natural cover or for semen collection and artificial insemination. Stud farm veterinarians should be aware of the techniques used to evaluate testicular function and the diagnostic approach to testicular disorders in cases of emergency. This paper presents the clinical methods used to evaluate testicular health, including palpation, ultrasonography, biopsy, and fine-needle aspiration. The discussion of testicular disorders is broken down into four categories: congenital disorders (cryptorchidism, monorchidism, and testicular hypoplasia), differential diagnosis of scrotal enlargement, differential diagnosis of causes of progressive testicular enlargement, and differential diagnosis of testicular asymmetry or reduction in size with an emphasis on testicular degeneration. The sudden increase in testicular size is often accompanied by severe clinical signs and is a major cause for referral of stallion for surgery. Testicular disorders are illustrated with clinical cases seen by the authors.

## 1. Introduction

Normal testicular function is paramount for breeding efficiency and determination of stallion book during the season [1,2,3]. Inherent fertility and quality of management affect reproductive efficiency, particularly in natural mating systems [4]. Testicular disorders are identified during neonatal examination, selection of prospective stallions for breeding, examination for subfertility, or medical emergencies. Disorders encountered during neonatal examination are primarily congenital, raising the question of heritability. In many countries, congenital defects prohibit the use of stallions for reproduction. Breeding soundness examination of a stallion requires a systematic evaluation of its health, followed by reproductive system examination, including evaluation of the scrotum and its content and semen collection and evaluation. Testicular health and spermatogenic efficiency are essential to stallion breeding soundness [5,6,7]. Acute testicular disorders should be considered as differential diagnoses for emergencies of breeding stallions presenting with colic, fever, or unexplained lameness.

This paper aims to review the diagnostic approach to testicular disorders in stallions. A detailed discussion of the techniques used to evaluate the testes is provided, followed by a review of the clinical presentation and illustration of the major categories of testicular disorders. Testicular disorders are broken down into those of congenital origin, those causing acute scrotal/testicular enlargement, and those causing progressive testicular enlargement asymmetry. Testicular degeneration, a common disorder in subfertile or infertile stallions, is discussed separately.

## 2. Examination Techniques of the Stallion Scrotum and Its Content

### 2.1. History and Physical Examination

A thorough history is vital to breeding soundness examination and helps determine the surrounding events that led to subfertility. Testicular disorders are often an emergency, and they can be associated with breeding accidents, trauma (hemorrhages), and, less commonly, a systemic or local infection (orchitis, epididymitis). An increase in scrotal size is more common in the cases of testicular tumors or certain circulatory disorders. Some testicular tumors may not be noticed until there is a sudden appearance of an intense local reaction that is usually accompanied by hydrocele or local edema. Four major scenarios can be encountered in practice in cases of testicular disorders with or without asymmetry of the scrotum: sudden enlargement with clinical signs dominated by intense pain, sudden enlargement without other clinical signs, a progressive increase in the scrotal size with development of an acute episode due to complications, or incidental finding following a reduction in fertility.

Regardless of complaint, all stallions should undergo a complete physical examination before the examination of the scrotum and its content. For this examination, the stallion should be sedated and placed in stocks. The authors prefer to sedate the horse with detomidine (0.01–0.02 mg/k, IV) and butorphanol (0.01–0.02 mg/kg, IV). This combination provides a good sedation depth and provides adequate time for examination. Spontaneous ejaculation may occur once the stallion is sedated. Therefore, semen collection for evaluation should be performed before performing this examination unless it is an emergency. Certain disorders, such as acute orchitis or epididymitis, inguinal/scrotal hernias with compromised strangulating bowels, spermatic cord torsion (>360 degrees), or rupture of the albuginea may cause severe pain. In these cases, a systematic approach to diagnosing a colicky syndrome should be used to rule out other causes of colic (i.e., gastrointestinal or urinary origin).

The general clinical approach to assessing the testes includes external inspection and palpation of the scrotum, followed by percutaneous ultrasonography. Semen collection and evaluation should be part of the evaluation of testicular function except in case of emergency. An important part of the breeding soundness examination is addressing the serological status of the major infectious diseases, such as viral arteritis, infectious anemia, equine influenza, and, less frequently, Dourine, as these diseases may cause sudden scrotal enlargement or edema [8,9].

### 2.2. Testicular Inspection, Palpation, and Measurements

Testes should be freely movable within the scrotum without signs of discomfort on palpation. Direct palpation of the testes often provides sufficient information when there is an acute injury. Scrotal wall thickness may indicate a chronicity of the lesion. Testicular size, shape, consistency, and temperature should be assessed. Testicular measurements, length, width, and height can be taken with calipers or ultrasonography. The measurements taken, include the total scrotal width (Figure 1) and the length, height, and width of each testicle (Figure 2).

These measurements help us to calculate the total volume of the testes and provide estimates for the daily sperm output based on established formulas (Table 1) [10].

Testicular measurements are taken either with calipers or by ultrasonography (Figure 3). Ultrasound techniques are more precise, as they measure only the testicular parenchyma, avoiding the layers of scrotal skin that are normally included in caliper measurements [10]. As some testes do not have an ellipsoid shape, obtaining images through 3D ultrasonography seems more precise, helping to avoid miscalculating the daily sperm output of a particular animal [11]. However, 3D ultrasonographic techniques are not commonly used in practice.

### 2.3. B-Mode Ultrasonography

Ultrasonography is an important part of testicular evaluation [12], which not only allows us to determine the size of the testes, but also allows us to evaluate the parenchyma and the vaginal cavity for any abnormality [13]. A multifrequency linear transducer (5–7.5 MHz) is preferable to assess the testes, epididymides, and spermatic cords [14]. This provides information about the anatomic structure and the echotexture of the testicular parenchyma (Figure 4) [15]. Each testicle should be scanned in the transverse and longitudinal planes, and testicular echotexture should be compared with the contralateral testis [14].

Several pathological conditions can be identified through ultrasonography, such as accumulation of fluid content within the vaginal cavity (hydrocele, hematocele, pyocele) [16], the presence of gastrointestinal organs (small intestine, omentum) [17], neoplasia (seminoma, Sertoli cell tumor, interstitial cell tumor, etc.) [18] thickening of the vaginal, parietal, or albuginea tunics, an increase in the size of testicular-related anatomic structures (spermatic cord torsion, epididymitis, orchitis, abscesses). In severe cases of testicular degeneration, there are marked changes in the echogenicity of testicular parenchyma, and these changes are either focal or disseminated. However, during early onset, the condition is difficult to differentiate from that of a normal testicle [5].

Recently, a computer echotexture analysis was proposed as an alternative non-invasive method with which to assess organ functionality. The analysis is based on detecting different pixel intensities and pixel heterogeneity on ultrasonographic images from testicular parenchyma [19]. In a testicular degeneration model, acute degenerative changes were reflected by increased pixel intensity in beef bulls [20]. In stallions, the pixel intensity of the testes does not seem to reflect the degree of testicular fibrosis [19].

### 2.4. Doppler Ultrasonography

Testicular function is highly dependent on proper testicular perfusion. Hemodynamic disturbances are the most common causes of subfertility [21]. This technique is a good early indicator of acute pathologies related to vascular disorders. Imaging of testicular vasculature is essential for distinguishing between spermatic cord torsion (lack of blood flow) and epididymitis/orchitis (increased blood flow) [22]. It is more reliable for visualizing short fragments of the longitudinal sections of the lumen of the testicular artery than gray-scale ultrasonography. Differences in the vascularization patterns of testicular tumors and abnormal course of large blood vessels within the testicular parenchyma after trauma are also identified using Doppler ultrasonography [21].

### 2.5. Fine-Needle Aspiration

Fine-needle aspiration (FNA) is considered to be a safe, minimally invasive, rapid, and inexpensive technique that allows sampling of the testicular parenchyma at multiple sites. Cytological evaluation of the samples provides an evaluation of the spermatogenic activity based on the proportion of different cells [23,24] (Table 2). This technique has been used to monitor spermatogenic activity in case of unexplained infertility [12], following a testicular injury/insult [24], and for the diagnosis of testicular neoplasia [25].

The stallion is sedated and restrained in stock. The scrotal skin is prepared aseptically. The testis is firmly held down within the scrotum, and then a sterile needle (22–23G × 1 ¼”) attached to a 10 mL syringe is repeatedly introduced into the testicular parenchyma at varying angles while drawing the plunger. It is important to release the plunger before withdrawing the needle; otherwise, the aspirate is aspirated into the syringe and cannot be recovered. Hemorrhage at the insertion site may be controlled by direct digital pressure. The material contained within the needle is later expelled on a glass slide to allow fixation and further staining with Giemsa [24]. The sample is evaluated at high magnification (×1000), and the proportion of cell type is reported (Table 2). Leydig cells are difficult to recover and are not easily recognizable in smears [24,26].

### 2.6. Testicular Biopsy

Testicular biopsy is not routinely used to aid in the diagnosis of subfertility because of the risks of complications if the procedure is not performed appropriately [27]. The primary indication is to investigate azoospermia/oligozoospermia [28].

Different biopsy techniques have been used in stallions, such as the open method, split-needle biopsy, and needle-punch biopsy. The size of the biopsy instrument used should be chosen according to the size of the testis, and it varies between the recommended 14 g for adult stallions [28] to 18 g for miniature stallions [29]. The most commonly used method is the split-needle biopsy (Biopty, Tru-Cut) [28] (Figure 5). Because of the high vascular component of testes, color Doppler ultrasonography can help to identify a good puncture site prior to the procedure [29]. The stallion is sedated and restrained in a stock. The scrotum is aseptically prepared, and local anesthetic is administered subcutaneously at the site of biopsy. The testis is held down firmly, and a small stab incision is made over the scrotal skin. The authors prefer the use of a self-firing biopsy needle. The needle is placed against the albuginea and fired (Figure 5) [28]. Firm compression with sterile gauze for 5 min provides adequate hemostasis. The sample obtained may not be sufficient for a detailed quantitative evaluation of spermatogenesis, and taking more than one sample without removing the needle is advised [29]. The recovered material can also be used to analyze abnormalities restricted to gene expression [30] and cDNA mutations analysis in subfertile stallions [31].

The procedure is relatively safe and rarely causes serious complications in stallions [28]. No significant side effects were found following multiple biopsies from the same stallions [27,29].

### 2.7. Semen Collection and Evaluation

Semen evaluation provides valuable information on spermatogenesis, sperm maturation, response to treatment, and prognosis for fertility [32,33]. Assessment of motility and concentration (azoospermia, oligozoospermia) of spermatozoa, along with determination of morphological abnormalities (teratozoospermia) and the presence of other cells such as large round cells (spheroids), medusas, and white blood cells should be performed [34].

### 2.8. Endocrinological Testing

Measuring the serum concentration of reproductive hormones is useful for determining the prognosis of subfertile or infertile stallions [35]. However, high seasonal variations in plasma hormone concentrations make interpretation of the results difficult [36]. The most commonly measured hormones are testosterone, inhibin, estrogens, and anti-Müllerian hormone (AMH). Follicle-stimulating hormone (FSH) and Luteinizing hormone (LH) are not routinely measured and only a few laboratories perform these assays [37].

Severe cases of testicular degeneration commonly present with increased concentrations of FSH and LH and decreased concentrations of estradiol and inhibin. These hormonal changes are identical to the hormonal changes in stallions with idiopathic infertility [37,38]. A decrease in serum testosterone concentration is observed as a sign of more advanced testicular degeneration.

### 2.9. Thermography

Thermography is a non-invasive technique that is used to measure the difference in temperature between the surface of the scrotum and underlying tissue structures. In fertile stallions, quantification of the thermic balance before and after semen collection has been proposed as a method to assess sperm production potential [39]. Thermography use may have an application in cases of scrotal disorders, such as trauma or testicular degeneration. Appropriate environmental conditions must be addressed prior to the procedure, as false high-temperature readings may occur [40].

## 3. Congenital Abnormalities of the Testis

### 3.1. Monorchidism

The term monorchidism has been used loosely in the literature. The strict definition of monorchidism is complete agenesis of a testis. This is a very rare condition. It is sometimes confused with severe degeneration of a cryptorchid testis caused by ischemic necrosis [41,42,43]. A few cases of monorchidism have been reported in various horse breeds: Thoroughbred, Appaloosa, Arabian, Brazilian horse, and Shetland pony [41,44,45,46]. These cases are often presented for suspected cryptorchidism. Monorchidism with an intraabdominal retained testis was reported in a 64, XY, SRY-positive quarter horse with a female phenotype. The diagnosis was confirmed by endocrine testing (testosterone, AMH, and Inhibin) pre- and post-surgical removal of the abdominal testis [47].

### 3.2. Testicular Hypoplasia

Congenital testicular hypoplasia has been described in several cases. It is characterized by a smaller-than-normal testicular size, and by oligozoospermia or azoospermia. Testicular hypoplasia has been associated with various cytogenetic abnormalities (XX/XXY, XY/XXY, XX/XY/XXY, XXY/XXYY). Testicular hypoplasia and sterility due to lack of spermatogenesis have been described in two stallions (a French trotter and an American standardbred) with 65XXY-syndrome [48]. Poor testicular development and spermatogenic arrest (Figure 6) were described in a sterile azoospermic Friesian stallion with a reciprocal translocation rcp (13; Y) (pter; qter) [49]. Clinically, in the authors’ experience, the testes may be softer or harder than normal, and testicular measurements are often significantly smaller than expected. Libido is often not affected by testicular hypoplasia.

### 3.3. Congenital Inguinal/Scrotal Hernia

Congenital inguinal hernias involve the ileum, distal jejunum and, occasionally, the large colon [50,51]. Indirect inguinal hernia (i.e., herniated content passing through the vaginal ring into the vaginal process) is the most common [51]. The hernia is often discovered during neonatal examination. The scrotum is visibly swollen, and the herniated tissue can be reduced by palpation (Figure 7).

Congenital hernias can be unilateral or bilateral. Generally, the foal does not show clinical symptoms (i.e., abdominal pain) unless the herniated organs become incarcerated or ruptured [52,53,54]. Indirect reducible congenital inguinal hernias can be managed conservatively (with repeated manual reduction or truss application after manual reduction) (Figure 8). Surgical management is indicated in cases of ruptured or nonreducible hernias. Traditionally, surgical repair of inguinal hernia included castration [54]. More recently, various non-invasive (laparoscopic) approaches have been described, which can be used to close the vaginal ring with or without castration [50,55,56,57]. However, using horses that underwent congenital inguinal hernia repair for breeding is debatable, and the condition is presumed to be hereditary [55].

### 3.4. Cryptorchidism

Cryptorchidism is the most common disorder of sexual development in horses, with a prevalence of 2 to 12%. Some breeds (Quarter horse, Saddlebred, Percherons, Friesian, Swedish Icelandic, and Mangalarga) are more represented than others. The incidence of cryptorchidism in a herd of Arabians was 2.5% [58]. The pathophysiology of cryptorchidism has yet to be completely understood [59]. In normal colts, testicular descent occurs between 30 days prepartum and the first 10 days after birth. Testicular descent involves hormonal and physical mechanisms, and its failure is multifactorial [59]. Recent studies have shown that genetics play a role in some populations of horses (Friesians and Swedish-born Icelandic horses) [60,61]. Due to the genetic component, the *Stud Book* rules of most of the breeds exclude cryptorchid stallions from breeding stock. However, the heritability of the condition is questioned by other authors. Cryptorchidism is commonly seen alongside sex-reversal syndrome [58].

Cryptorchidism is classified based on the location of the retained testis as (1) complete abdominal (testis and epididymis entirely within the abdomen), (2) partial abdominal (abdominal testis and inguinal epididymis), and (3) inguinal (testis and epididymis within the inguinal canal). Most cryptorchidism cases are unilateral (85–90%), involving the left and right testis equally (see review [58]). However, a predisposition of one side over the other may occur in some breeds and for some ages. The right testis was significantly more retained than the left in Friesians [61] and in young ponies [62]. Various neoplastic changes (seminoma, teratoma, carcinoma, interstitial cell tumors, malignant Sertoli cell tumors, and leiomyosarcoma) have been reported in abdominally retained testis (see review [58]). Some of these tumors can be associated with recurrent colic due to torsion or jejunal entrapment and strangulation [63,64].

Diagnosing cryptorchidism is straightforward in uncastrated horses with a precise history. The diagnosis becomes challenging when there is no precise castration history or when horses presumed to be castrated present stallion-like behavior. In horses with no evidence of descended testicles, several endocrine tests are available to determine the presence of testicular tissue. Determination of serum testosterone levels before and two hours or 24 h after hCG administration (10,000 IU, IV) improves the accuracy of diagnosis, but about 6.7% of the tests are inconclusive. The determination of serum estrogen levels is inaccurate for horses younger than three years and is subject to seasonal variations. Serum AMH level is an excellent marker for the presence of Sertoli cells. It is more reliable for detecting the presence of testicular tissue in horses without a scrotal testis (Table 3) [65]. False negatives have been reported and are attributed to insufficient number or degeneration of Sertoli cells [41].

Although the diagnosis of cryptorchidism can be established by endocrine testing, clinical evaluation and the determination of the location of the retained testis are required before surgery [58,66]. This allows the evaluation of testicular tissue for any pathology and the selection of the best surgical approach based on location. Clinical evaluation is performed after sedation of the horse. Evaluation of the suspected cryptorchid horse starts with palpation of the inguinal area. However, inguinal palpation is not sufficiently sensitive and allows the identification of only 60% of the superficially retained testes and 7% of the deep inguinal testes [58]. The combination of transcutaneous inguinal and abdominal ultrasonography allows the determination of the location of the retained testis with higher sensitivity (97.6%) and specificity (100%) [58,66,67,68] (Figure 9, Figure 10 and Figure 11). Examination of the horse after 24 to 36 h of fasting improves visualization of the abdominal testis. Transrectal ultrasonography, when possible, allows better visualization of the abdominal testis. After sedation, the internal genitalia is examined transrectally and the ampulla of the vas deferens is identified and followed on each side toward the inguinal area (Figure 12) [58,69,70].

Failure to locate the retained testicles despite using a combination of palpation, transcutaneous, and transrectal ultrasonography is possible when there is true monorchidism or severe ischemic necrosis of the retained testis [58].

### 3.5. Other Congenital Abnormalities of the Testis and Epididymis

Other congenital abnormalities of the testis reported in stallions include cystic rete testis in cryptorchid horses [71,72], and epididymal or ductus deferens segmental aplasia [73,74]. However, compared to cryptorchidism and testicular hypoplasia, these disorders remain very rare.

## 4. Differential Diagnosis of Acute Scrotal Enlargement

Scrotal enlargement is a common complaint and is considered to be an emergency in equine theriogenology practice [75,76,77]. The enlargement can be caused by injuries that occur during breeding, while jumping fences, or as a result of other accidents [76,77,78]. Scrotal/testicular enlargement can be acute or progressive. Acute scrotal enlargement is generally caused by accidents or trauma (testicular hemorrhage, inguinal hernia, spermatic cord torsion) or by a systemic or local infection (orchitis, epididymitis) [76,79]. These conditions lead to a painful scrotum with increased sensitivity to touch and sometimes even a colicky syndrome [75,80]. Such conditions should be considered an emergency, sometimes requiring referral to a specialized hospital [76]. Clinical management requires a specific diagnosis and involves history, physical examination, and specific examination of the scrotum, spermatic cord, testes, and epididymides. Hematology and blood chemistry are indicated in cases with systemic involvement and fever. Initial management includes pain management, commonly followed by hemicastration to preserve the fertility of the stallion [77]. The immediate treatment involves cold application (hydrotherapy or a sling to hold a cold pack) and therapy with NSAIDs (flunixin meglumine, 1.1 mg/kg, intravenously [IV] or orally or phenylbutazone, 4.4 mg/kg, IV or orally) to reduce pain and inflammatory mediators. Sedation is indicated in most cases [81]. The use of antibiotics, topical wound treatment, and diuretics is indicated in traumatic non-surgical cases [77,82].

### 4.1. Scrotal and Testicular Trauma

Scrotal trauma results in inflammation, hemorrhage, edema, contusion, or testicular rupture [76,77,83] (Figure 13). Open scrotal lesions may lead to bacterial infection [76]. Although edema can be diagnosed on visual examination of the scrotum, more precise detection of the tissue insult requires palpation and ultrasonography. A thickened scrotal wall suggests local tissue damage or generalized edema of the scrotum (Figure 14) [80].

The rupture of tunica albuginea and hemorrhage makes the scrotum too firm for the testes to be palpated. The severity of tissue damage may range from edema to rupture of tunica albuginea and hematocele [78,84]. On ultrasonography, hematocele and testicular hemorrhage are anechogenic until clot formation. Contrarily, scrotal edema and induration present a heterogeneous echotexture [76]. The ultrasonographic appearance of testicular hemorrhage (Figure 15a) becomes more heterogeneous with the organization of the blood clot and shows echogenic areas corresponding to fibrin [76]. Although scrotal ultrasonography and tissue echogenicity may help differentiate between scrotal edema and testicular hemorrhage, they are not always diagnostic. In such cases, aseptic fine-needle aspiration helps to confirm the diagnosis [76]. However, before aspiration, inguinal hernia must be ruled out by external and internal inguinal rings palpation and ultrasonography. Torsion of the spermatic cord and scrotal edema can be differentiated by ultrasonography. Intrascrotal hemorrhage from a ruptured spermatic cord or tunica albuginea can be complicated by the formation of adhesions between the scrotal fascia and the vaginal cavity leading to immobilization of the testicle inside the scrotum [85]. Rupture of the tunica albuginea after testicular trauma cannot be reliably visualized by ultrasonography. However, it may be suspected by the presence of herniation of the epididymis and the testicular parenchyma [76].

Trauma leading to testicular rupture (Figure 15b) and hemoperitoneum has been reported in a 13-year-old American Quarter Horse. In this case, the diagnosis was incorrectly made for inguinoscrotal hernia, highlighting the need for differential diagnosis of acute scrotal enlargement [78]. The clinical picture included colic with elevated heart and respiratory rates, hyperemic mucus membranes, increased capillary refill time, and decreased borborygmi [78]. The diagnosis was confirmed following exploratory laparotomy, and the stallion recovered after unilateral orchidectomy.

### 4.2. Acquired Scrotal/Inguinal Hernia

Inguinal hernias are often reported after trauma or exercise [17,76]. However, these factors are not obligatory in the etiopathogenesis of inguinal hernias. Some breeds (i.e., large-gaited breeds) and horses with large vaginal rings are more predisposed to inguinal hernias. The acquired hernia is often unilateral, but a bilateral hernia was described in a 7-year-old show-jumping stallion [85]. The hernia can be direct or indirect inguinal or scrotal. The differential diagnosis is challenging [86]. Direct inguinal or scrotal hernia occurs when the intestines enter the inguinal canal (between the vaginal ring and the external inguinal ring) through a breach in the peritoneum and transverse fascia adjacent to the inguinal ring. In indirect hernia, the intestines pass through the vaginal ring into the scrotum or strangle within the inguinal canal. Indirect inguinal hernias are more common than direct hernias [86].

In stallions, inguinal hernias mainly involve the passage of part of the jejunum or the ileum. However, herniation of the large intestine, omentum, or urinary bladder has also been reported (Figure 16) [17,87,88]. These structures can be visualized within the vaginal cavity and evaluated for viability by ultrasonography (Figure 17). Acquired inguinal hernia with concurrent nephrosplenic entrapment has been reported in a 6-year-old Standardbred stallion [81]. An indirect acquired inguinoscrotal hernia of impacted and twisted pelvic flexure has been reported in a 3-year-old Andalusian stallion [88].

In the case of uncomplicated inguinal hernias, the only clinical sign is a discreet increase in the size of the scrotum, which is associated with slight preputial and ventral edema and sometimes reduced or absent borborygmi [89]. The stallion’s attitude may range from lethargic to showing signs of anxiety and pain, making transrectal palpation virtually impossible [76]. Early inguinal hernias are easily reduced by external or transrectal palpation. In advanced cases, other symptoms may appear, including depression, dehydration, increased heart rate (>70/min) and respiratory rate (>30/min), fever, hyperemic mucous membranes, and increased capillary filling time [76,81,86]. The state of dehydration is variable, as indicated by the variability of hematocrit and plasma protein levels. The blood pH varies from normal to acidic [90]. The hematological examination often shows leukocytosis and neutrophilia. However, such hematological changes may be absent or slight, indicating a minimal impairment of homeostasis. The peritoneal fluid is transudative with high protein and leukocyte values or sometimes serous (modified transudate with low cellularity) [76]. When there is inguinal hernia or fluid accumulation in the vaginal cavity (hydrocele, hematocele, pyocele), the intrascrotal tissues surrounding the testes are soft on palpation. A thickened scrotal wall is suggestive of local tissue damage or generalized edema of the scrotum [80].

Intense clinical signs are common when the intestines are strangled, resulting in an emergency surgical intervention [85,86]. Acquired inguinoscrotal hernias represent 3–7% of the causes of colic in stallions [17]. However, in a recent study [91], the overall prevalence of acquired inguinal hernia was 1.6% of all horse colic cases, while surgery for acquired inguinal hernia represented 25% of the surgeries in stallion due to colic. The risk of acquired inguinal hernia increases in late spring and summer because those are the times in which horses are competing/exercising/breeding. Acquired inguinal hernia results in poor short-term survival for the draft breed, in cases presented with elevated heart rate, and in the presence of small intestine volvulus [91]. The cases presented within 10 h of the onset of clinical signs (colic) have a significantly higher survival rate than those presented later [91]. Acquired inguinoscrotal hernia had an 11% reoccurrence rate in a study where 65% of cases were treated surgically and 33% reduced manually. The reoccurrence was seen in the cases treated by manual reduction or surgically (with bilateral castration, unilateral castration, or without castration) including bilateral standing laparoscopic procedures [91].

Castration and closure of the vaginal ring is the treatment of choice to avoid recurrence in cases of inguinoscrotal hernia [91]. In cases of incarcerated inguinal hernia, surgical correction can be attempted through combined ventral midline celiotomy and an inguinal approach [92]. In early cases with healthy intestines, a hernia reduction followed by a reduction in the size of the vaginal ring via laparoscopy without castration is an alternative approach [50,52,56].

### 4.3. Torsion of the Spermatic Cord

Torsion of the spermatic cord, incorrectly called testicular torsion, is the rotation of the spermatic cord along its vertical axis [76,93]. It is a surgical emergency [93,94]. The vertical relation between the longitudinal axis of the spermatic cord and testis increases the risk for spermatic cord torsion [93]. Torsions of 180° or less are more common incidental findings and do not cause any clinical signs [76]. The diagnosis of 180° torsion is based on the position of the tail of the epididymis and the proper ligament of the epididymis (Figure 18). Torsions of more than 180° result in clinical changes in scrotal contents mainly due to occlusion of the blood flow to the testis, leading to tissue hypoxia, ischemia, free radical production, thrombosis, and apoptosis of germ cells [93]. The diagnosis of severe torsion is based on a good history and a detailed examination of the scrotal contents and the inguinal region. The cauda epididymis position does not help to diagnose 360° spermatic cord torsion [82], but it may be palpable in the cranial aspect of the testis in cases of 720° torsion [95]. However, in most cases, the scrotum is too tense and edematous, and palpation of the cauda epididymis is not possible.

Spermatic cord torsion must be suspected in cases of acute colic and enlarged painful scrotum, scrotal and preputial edema, and thickened spermatic cord [76,80]. Differential diagnoses include other conditions causing an enlarged painful scrotum, such as inguinal/scrotal hernia, scrotal trauma, orchitis/periorchitis, certain tumors, epididymitis, testicular hemorrhage, and thrombosis of testicular cord vessels [76,96]. Ultrasonography of the scrotal contents and palpation of the vaginal ring on the affected side makes it possible to establish a differential diagnosis. On ultrasonography, the testicular parenchyma is irregular and heterogeneous, and the spermatic cord has decreased blood flow (Figure 19) [82]. Edema of the scrotum and testis can also be evident on ultrasonography in some cases [95]. In untreated cases, the testicular parenchyma shows increasingly irregular echogenicity within 36 h (Figure 19) [82]. The integrity of the arterial supply (testicular artery) can be examined by color Doppler or power Doppler ultrasonography [21,76,96]. Stallions with 180° spermatic cord torsion may have a retrograde diastolic blood flow [21]. Ischemia and oxidative injury follows the decrease in the blood perfusion of the affected testis [95].

Orchiopexy is recommended in human spermatic cord torsion cases but is not performed in stallions. Hemicastration is the treatment of choice alongside postoperative antibiotics and pain management [82,95]. Bilateral castration is recommended for horses that are not destined for breeding [95]. Histological evaluation would reveal ischemic (coagulative) necrosis of the affected spermatic cord and testis [82]. The torsion of the remaining spermatic cord has been reported in a 5-year-old stallion two years after hemicastration, and the torsion was surgically corrected and sutured in place [97]. Normal conception rates are expected following hemicastration as long as the remaining testicle is normal [97].

### 4.4. Orchitis and Epididymitis

Orchitis and epididymitis are rare in stallions [16,98,99]. Orchitis can be caused by infections extending from a scrotal wound, hematogenous spread, trauma, parasites, and extension of infection from the accessory sex glands [16,78,98,100]. Immune-mediated orchitis is caused by traumas resulting in testicular rupture and exposure of germ cells to the circulation [77]. The testis is typically enlarged, hot, and painful. Lameness, and systemic signs of fever, anorexia, and listlessness, may be present is some cases [16,79,98,101]. Hematology may reveal neutrophilia, leukocytosis, anemia, elevations of liver enzymes, and hyperproteinemia with mild hypoalbuminemia and marked hyperglobulinemia [16,101]. Bilateral orchitis with unilateral inguinal hernia has been reported in a seven-year-old stallion [102].

Bacterial orchitis and periorchitis are rare. The most common isolates are *Streptococcus equi* ssp. zooepidemicus, *Actinobacillus equuli*, *Pseudomonas mallei*, *Salmonella abortus* equi, and *Escherichia coli*. Corynebacterium pseudotuberculosis was reported in unilateral orchitis (Figure 20) and epididymitis [16,103]. Epididymitis is usually of infectious cause in stallions and reported pathogens include Streptococcus zooepidemicus and Proteus mirablis [98]. Strongylus and Setaria genera of nematodes can migrate to the testes and cause can cause orchitis and epididymitis [104]. Enlarged epididymides due to migration of neoplastic cells were diagnosed in a 12-year-old Thoroughbred stallion suffering from generalized lymphosarcoma [105].

Orchitis and epididymitis must be differentiated from torsion of the spermatic cord, inguinal herniation, testicular neoplasia, hematocele, testicular trauma, and scrotal edema and hematoma [98]. The diagnostic approach includes ultrasonography of the scrotum and its contents, transrectal palpation, and ultrasonography [16]. The testes may be painful, hot, edematous, and enlarged. Ultrasonography may show anechoic fluid with irregular edges in the vaginal tunic (Figure 20), consistent with transudate, although heteroechoic abscess may be present in some cases [101,106]. However, ultrasonography may not differentiate between orchitis, testicular abscess, organizing hematoma or other causes of heterogenous echogenicity of testes [76]. A hematoma of the pampiniform plexus has been reported in a 6-year-old Crioulo stallion with an orchitis [101]. Periorchitis may sometimes lead to peritonitis [106]. Orchidectomy and/or hemicastration along with antibiotics and analgesics are the treatments of choice. The closure of inguinal rings is suggested to prevent inguinal hernia [106]. Good fertility has been reported in stallions with unilateral testicular degeneration (testicular asymmetry) following orchitis. The histopathology showed atrophy of seminiferous tubules, parenchymal sclerosis, focal calcification, and chronic macrophagic and lymphocytic inflammation [107].

In cases of epididymitis, abnormally dilated ductal segments are seen within the affected cauda epididymis (Figure 21) in the absence of other causes of testicular enlargement [108,109]. The head or tail of the affected epididymis is enlarged with hypoechoic areas of edema and inflammation [109]. The semen cytology reveals non-degenerative neutrophils and mononuclear cells in a stallion diagnosed with bilateral epididymitis [108].

## 5. Differential Diagnosis of Progressive Scrotal Enlargement

The causes of progressive scrotal enlargement include circulatory disorders such as hydrocele, pyocele, varicocele, and testicular tumors, although some testicular tumors are not noted until acute clinical signs are presented [76].

### 5.1. Hydrocele

Hydrocele is an accumulation of fluid within the vaginal cavity. The scrotal sac has the consistency of a fluid-filled squeeze bag [76]. Hydrocele is often seen in high-heat and -humidity environments because of disturbance of the spermatic cord hemodynamics and leakage of peritoneal fluid in the vaginal cavity [76]. Hydrocele is a common finding in acute testicular enlargement due to testicular trauma, inguinal hernia, testicular torsion, or other conditions causing a decreased venous or lymphatic outflow [110]. *Strongylus* and *Setaria* genera of nematodes can migrate to the testes, causing an inflammatory reaction and hydrocele [104]. Mild hydrocele can be an incidental finding and is common in stallions during hot weather and at the end of the breeding season, especially after intense sexual activity, and it does affect fertility [110]. A mild-to-moderate hydrocele does not significantly affect vascular perfusion, although the scrotum will be enlarged (Figure 22) [76,110]. However, a large hydrocele can cause discomfort and, in severe cases, produces turbulent blood flow in the testicular artery at the level of the spermatic cord [21,76,110]. Ultrasonography can differentiate among the hydrocele, hematocele, pyocele, orchitis, epididymitis, scrotal edema and hemorrhage [76,83]. On ultrasonography, the hydrocele is characterized by anechoic or semi-echoic fluid within the vaginal cavity (Figure 22) surrounding the testicle and epididymis. In infectious or traumatic hydroceles, fibrin may also be present [76,110,111]. Some stallions are predisposed to developing large hydroceles in hot and humid environments. Cooling and regular exercise can reduce the incidence in these horses. Large hydroceles may be managed by aspiration of the fluid. In countries where the temperature and humidity can be very high, an air-conditioned stall is the only way to reduce their incidence.

### 5.2. Pyocele

Pyocele is the accumulation of pus within the vaginal cavity. The fluid appears hyperechoic on ultrasonography (Figure 23) [112]. Pyocele in a 9-year-old Andalusian stallion was characterized by increased testicular blood flow and bilateral enlargement of the spermatic cords and scrotum [112]. Pyocele, periorchitis, and suppurative orchitis leading to fatal peritonitis have been reported in a stallion [113]. The authors have seen cases of pyocele by extension of peritonitis following abdominal surgery (Figure 23).

### 5.3. Varicocele

Varicocele is an abnormal dilation of the veins (varicose veins) of the pampiniform plexus. It is suspected to be an underdiagnosed cause of subfertility in stallions [110]. The subfertility in varicocele is caused by compromised thermoregulation, tissue hypoxia or ischemia, and oxidative stress [114,115,116]. The pathophysiology of varicocele is not fully understood [117]. A high level of sperm membrane lipid peroxidation has been reported in stallions with varicocele [77]. In stallions, varicocele has been reported as a sequela of inguinal hernias, hydrocele, orchitis, and cardiac disease [110]. The spermatic cord and the dilated veins feel like a bag of soft tortuous strings on palpation. On ultrasonography, the veins appear dilated, irregular, anechoic, and as non-pulsatile structures of 8 to 24 mm in diameter (Figure 24) [118,119].

### 5.4. Scrotal and Testicular Tumors

Testicular tumors are rare, representing 0.04–0.9% of all equine tumors, and are generally unilateral [120,121]. They cause progressive, often painless scrotal enlargement [120,122]. However, in some cases, the tunica albuginea may rupture, causing intraparenchymal hemorrhage and a rapid increase in scrotal size [123], and edema and pain may develop quickly in the scrotal area, often resulting from inflammation rather than being directly caused by the tumor [76].

There are four categories of testicular tumors, depending on their origin: (1) germ cell tumors (seminoma, teratoma, teratocarcinoma, embryonal carcinoma), (2) sex-cord stromal tumors (Leydig cell tumors, Sertoli cell tumors, mixed sex-cord/stromal cells tumors), (3) mixed germ-cell sex-cord stromal tumors, and (4) primary tumors not specific to the testis (leiomyoma, leiomyosarcoma, lymphoma) [122]. The majority of testicular tumors are single and benign [120]. Most equine testicular tumors are of the germ-cell type. Seminomas are the most common in stallions and have a metastatic potential [121,122,124]. These are multilobular with gray or white surfaces (Figure 25).

Ultrasonography of the testes can reveal intratesticular or extratesticular masses that are not detectable on palpation. The tumors have lower echogenicity than the normal parenchymal tissue and may be solid or cystic (Figure 26). The ultrasonographic appearance ranges from circumscribed small nodules to large complex masses with disruption of normal testicular anatomy [18,83]. Seminomas appear as diffusely hypoechoic structures with ill-defined regions of hyperechogenicity, giving them an appearance of hypoechoic nodules within the testicular parenchyma (Figure 26) [122,125]. Confirmation of the diagnosis and determination of the tumor type is achieved by histopathology following testicular biopsy or castration [92]. On histology, seminomas show polyhedral tumor cells with large nuclei, prominent nucleoli, and a small border of cytoplasm (Figure 27) [121]. In the presence of metastasis from testicular tumors, lymphadenopathy can be palpated at the level of the medial iliac lymph node [76]. The superficial and deep inguinal lymph nodes can also be affected in the case of metastases [126].

Although rare in horses, Mesothelioma of tunica vaginalis in the region of pampiniform plexuses has recently been reported in a 2-year-old Standardbred stallion. The clinical signs included hemoperitoneum, anemia, and dependent edema [127]. Unilateral testicular mastocytoma resulting in progressive scrotal swelling and unilateral testicular enlargement was reported in a 12-year-old Peruvian Paso stallion [128]. Concurrent Sertoli cell and mixed sex cord-stromal tumors in a single descended testis have been reported in a 12-year-old Standardbred stallion [120]. Malignant testicular mixed sex cord-stromal tumor with metastasis to spleen, liver, lung, kidney, gastrointestinal tract, peritoneum, pericardium, pleura, bone marrow, and contralateral testis was reported in a 25-year-old stallion. The clinical signs were weight loss and progressive, painless unilateral scrotal swelling [129]. A malignant mixed sex cord-stromal tumor of dual Leydig and Sertoli cells was reported in a 30-year-old Standardbred stallion that presented with painless unilateral scrotal swelling [130]. Bilateral leiomyoma tunica albuginea causing progressive scrotal enlargement was diagnosed as multiple hyperechoic masses [131]. Simultaneous seminoma in the left testis, leiomyoma in the right testis and inguinal hernia with intestinal incarceration was diagnosed in a stallion presented for progressive scrotal enlargement and euthanized due to acute colic [132].

Moreover, progressive scrotal enlargement can be caused by scrotal skin tumors. The most common are melanomas and sarcoid. However, lymphosarcoma has also been reported in a Quarter horse stallion with nodular lesions in the scrotum (Figure 28) [76,133].

Testicular tumors must be differentiated from other causes of increased scrotal size, such as testicular torsion, orchitis, periorchitis, hydrocele, scrotal/testicular hematoma, testicular abscesses, spermatic granulomas and scrotal hernia by a detailed history and scrotal examination [83]. The history reveals a progressive increase in the size of the testicle (during the last six months) [76], although certain seminomas have been associated with a sudden increase in testicular size [134]. Acute colic and scrotal swelling due to Leydig cell tumor has been reported [135]. Inguinal or preputial masses may be observed in animals suffering from malignant tumors [63].

In the absence of metastasis, hemicastration is the treatment of choice in cases of testicular tumors. Compensatory hypertrophy of the contralateral testis may be observed following removal of the affected testis in younger stallions [23,120].

## 6. Asymmetric Testis/Testicular Degeneration

Because age-related changes are progressive, a thorough history will help differentiate idiopathic testicular degeneration from degeneration resulting from a testicular insult. Clinical signs such as small, soft testes may be more obvious in the advanced stages of testicular degeneration. Testicular degeneration is characterized by a subtle to marked reduction in fertility over time, with a unilateral or bilateral reduction in testicular size and daily sperm output. Due to the high demand for some stallions, a mild decrease in fertility can translate into significant economic losses. Idiopathic testicular degeneration is well documented in aged stallions, but its pathophysiology remains poorly understood [35]. There is no effective treatment for advanced cases of testicular degeneration in stallions [136]. Thus, early detection of testicular dysfunction is crucial for implementing a treatment and potentially recover the fertility of the affected animals [137]. Some stallions can be managed using assisted reproductive techniques, and their breeding life can thus be extended for some years.

A complete history is probably the most important component to aid in diagnosis. Breeding records may show a progressive reduction in fertility over the years. Due to the nature of the disorder, a decrease in pregnancy rate per cycle may go unnoticed initially. Any traumatic event or systemic disease may induce a transient reduction in fertility that may induce testicular degeneration if not addressed in a timely manner. Regular examination of the testes is important to detect major changes. Gray-scale ultrasonography allows the calculation of testicular volume and spermatogenic efficiency (DSO per milliliter of testis). However, testicular volume is often only affected in the later stages of the disease and no change in testicular echogenicity is visually detectable in the early stages [138].

Early diagnosis of testicular dysfunction due to vascular disturbance is crucial to address the future reproductive outcome of breeding stallions. Doppler ultrasonography can be used to aid traditional B-mode ultrasonography. Several Doppler parameters can be recorded, (resistive index, pulsatility index, Peak Systolic Velocity, End Diastolic Velocity, Time Average Maximum Velocity, Total Arterial Blood Flow, and TABF rate) [110], but their diagnostic value is not precise. The capsular artery seems to be the most reliable for Doppler assessment, since blood flow parameters of this artery are shown to be closely correlated with sperm-quality parameters [139,140]. Subfertile stallions tend to present high pulsatility and resistive index values (high vascular resistance). Clinical application benefits for suspected cases of testicular degeneration have not been reported. B-mode ultrasonography can be combined with computer-assisted pixel analysis of testicular parenchyma images [141]. The echotexture evaluation may increase the detection of minor changes in the histomorphology of the testes and, consequently, changes in semen quality [20,142,143]. Often, changes in the semen characteristics will be noticed before changes in testicular parenchyma are identifiable. To discover subtle changes in sperm quality, stallion sperm reserves must be depleted by one collection per day for one week. In progressive age-related testicular degeneration, some immature spermatogenic cells, commonly named “round cells” or multinucleated giant cells, may be present [36]. Samples should be stained with Giemsa or Diff-quick to differentiate these large cells from neutrophils and lymphocytes (Figure 29).

Assessment of proteomics and transcriptomics of testicular markers emerged as a new tool to understand the pathophysiology of testicular degeneration. Upregulation of cytokine-mediated inflammatory pathways in testes affected by degeneration seems to be part of the pathophysiologic events leading to the disease. Imbalances of growth factors, such as bone morphogenic protein 4 (BMP4) and cytokine-mediated inflammation (CSF1), lead to impaired response from stem spermatogonia cells (self-renewal vs. differentiation) may lead to germ cell loss [144]. Although low-grade inflammation may be involved, an early treatment has yet to be investigated. Protein SP22 has been suggested as a potential biomarker for fertility in humans. It is expressed in equine spermatozoa and participates in the level of interaction with oocytes. SP22 is incorporated into the spermatozoa during spermatogenesis [145]. Testicular degeneration alters SP22 expression in stallion spermatozoa [146]. However, this observation has yet to be standardized for testing breeding stallions.

Although hormonal assays can be performed in cases of suspected testicular degeneration, the results are often unrewarding. The most commonly measured reproductive hormones have markedly diurnal and seasonal variations, making interpretation difficult [35]. Elevated serum FSH and LH, low estradiol, and low testosterone are found in advanced cases of testicular degeneration. Often, once hormonal changes become apparent, the condition can be diagnosed by measuring testicular size and echogenicity and alterations in the ejaculated number of spermatozoa [35].

Serum AMH (anti-muller hormone) concentration increased rapidly in mice after the administration of various gonadotoxic chemotherapy drugs [147]. In stallions, AMH was proposed as a possible biomarker of testicular damage by toxic insult, based on the elevation of blood levels in those animals with testicular degeneration induced by the anti-spermatogenic/gonadotoxic compounds [148]. These changes may be due to a transient reversal of Sertoli cells to a prepuberal state, where secretion of AMH is elevated [149]. Other markers of Sertoli cells immaturity such as cytoskeletal proteins, including inhibin-alpha, were also found in canine testes showing signs of atrophy [150]. Monitoring changes in AMH may help with the early detection of testicular degeneration. The use of thermography relies on the principle that the inability to maintain testicular thermoregulation can result in acute testicular degeneration, which may evolve into chronic degeneration. Temperature variations between the two sides of the scrotum, along with a reduced scrotal skin temperature (SST) gradient from the base to the apex of the scrotum, and a 2–3 °C increment in SST may indicate the onset of testicular degeneration [39].

Fine-needle aspiration and testicular biopsy are indicated in cases of suspected azoospermia of testicular origin. When inflammation becomes persistent over time, the epithelia of the seminiferous tubules may begin to degenerate. An evaluation of the testicular cell population can provide information regarding the activity of the seminiferous tubules. The sample can be taken through FNA or via a biopsy. The histological evaluation can better show what is happening in the seminiferous tubules. It is used to substantiate the presence of testicular degeneration and to differentiate it from azoospermia due to occluding ejaculatory disorder [27]. The most common findings include cytoplasmic vacuolization, disorganization of the seminiferous tubule epithelia, and absence of some stages of differentiation of germ cells (Figure 30). In advanced cases, fibrous tissue and seminiferous tubules are devoid of spermatogenesis, with only Sertoli cells and some spermatogonia seen [144].

## 7. Conclusions

Stallion testicular disorders can be congenital, acute, or progressive changes. In this paper, we propose a systematic approach to diagnosing these disorders and relate the available literature on each. Clinical evaluation of young colts can detect all congenital defects. Cryptorchidism is the most common congenital defect of testicular development and a thorough transcutaneous and transrectal ultrasound examinations should allow a veterinarian to determine the locations of the retained testis. Traumatic and infectious testicular or scrotal insults are usually accompanied by a sudden increase in scrotal/testicular size with or without significant systemic involvement. These events represent a significant cause of colic in stallions. A precise diagnosis can be reached through a complete history and clinical evaluation. Testicular ultrasonography is an important clinical skill for evaluating all testicular disorders, especially in emergencies. Clinical evaluation of stallions presented for subfertility should include evaluation of testicular function through testicular measurements, ultrasonography, and semen collection and evaluation. Testicular biopsy or fine-needle aspiration may be helpful to refine the diagnosis. Idiopathic testicular degeneration remains one of the most important disorders in breeding stallions. However, its pathophysiology is not well understood. Future studies on proteomics may lead to the development of tests for early diagnosis of the disease.

## Figures and Tables

**Figure 1 vetsci-11-00243-f001:**
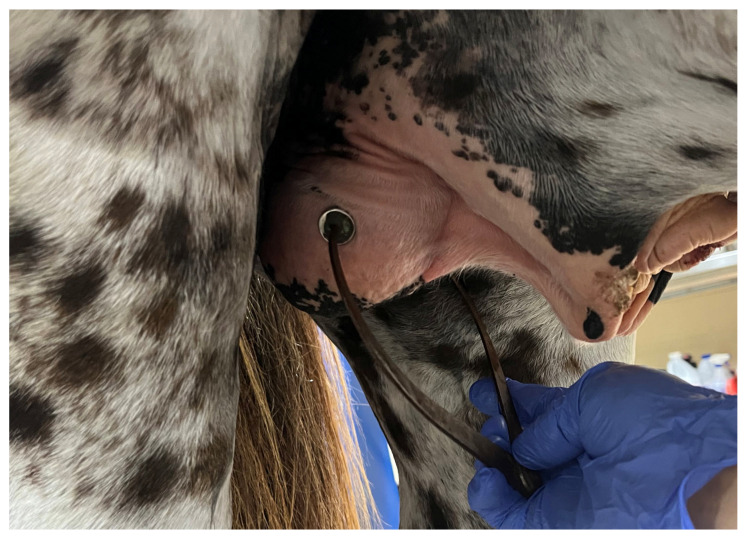
Measurement of the total scrotal width in a stallion using calipers.

**Figure 2 vetsci-11-00243-f002:**
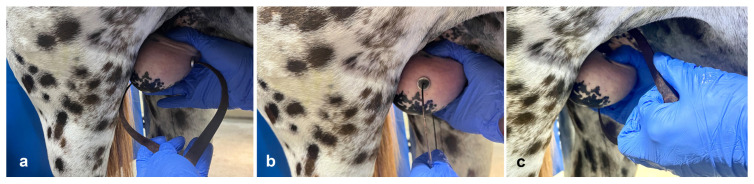
Testicular measurements using calipers technique. (**a**) Length, (**b**) width, (**c**) height.

**Figure 3 vetsci-11-00243-f003:**
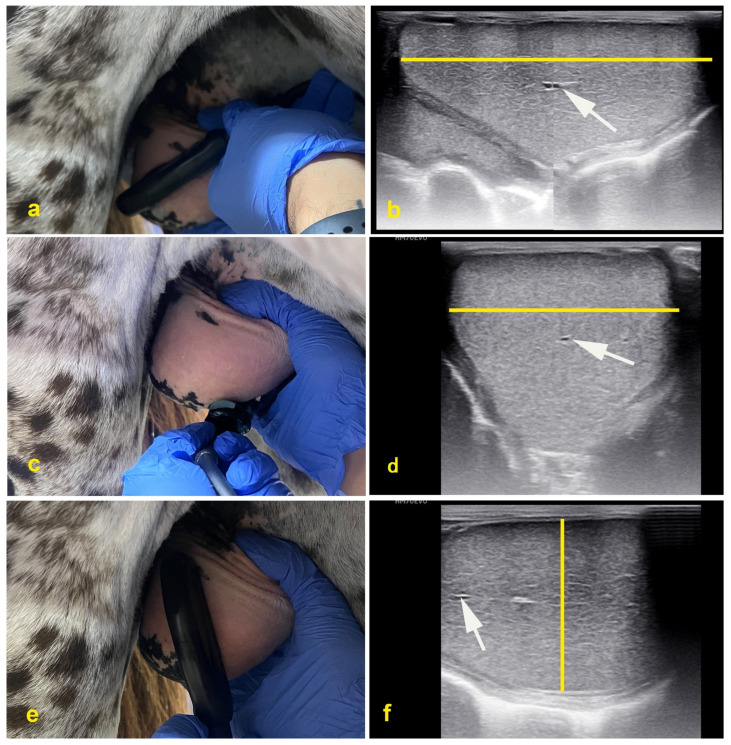
Testicular measurements by ultrasonography (**a**,**b**) length, (**c**,**d**) width, (**e**,**f**) height. Note the homogeneous appearance of testicular parenchyma. Arrows indicate the central vein (mediastinum testis).

**Figure 4 vetsci-11-00243-f004:**
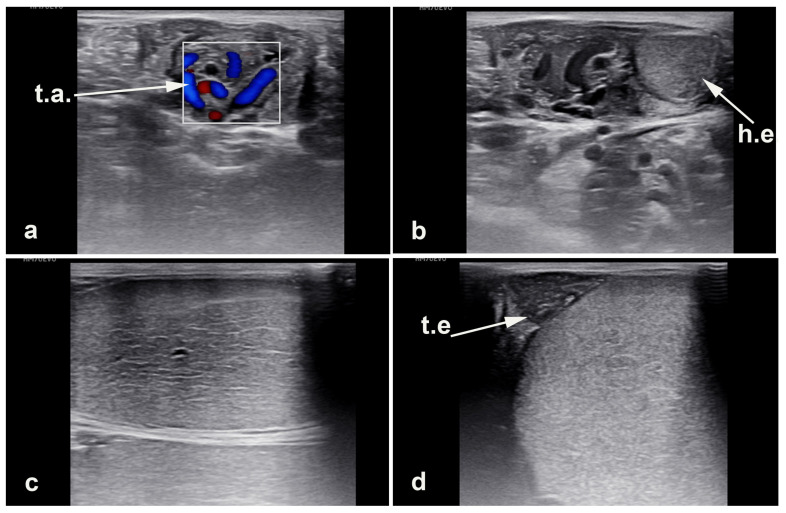
Normal testicular ultrasonography. (**a**) spermatic cord (t.a. testicular artery), (**b**) head of the epididymis (h.e), (**c**) testis, (**d**) tail of the epididymis (t.e).

**Figure 5 vetsci-11-00243-f005:**
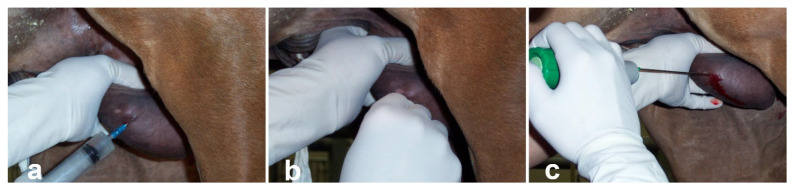
Testicular biopsy technique in stallions using a self-firing Bard Biopty instrument. (**a**) administration of lidocaine at the biopsy site, (**b**) incision of the scrotal skin and vaginal tunic, (**c**) insertion and firing of the biopsy instrument.

**Figure 6 vetsci-11-00243-f006:**
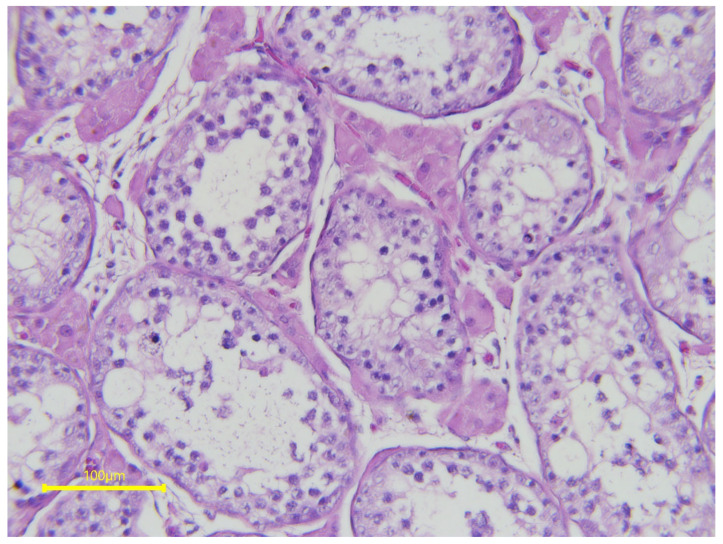
Testicular histology from a 5-year-old Friesian stallion with spermatogenic arrest and reciprocal translocation rcp (13; Y) (pter; qter).

**Figure 7 vetsci-11-00243-f007:**
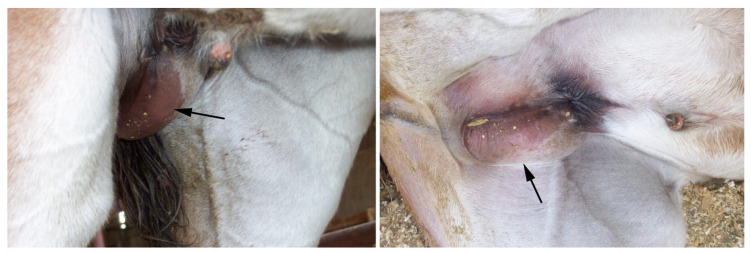
Congenital inguinal hernia in an 2-day-old Arabian foal.

**Figure 8 vetsci-11-00243-f008:**
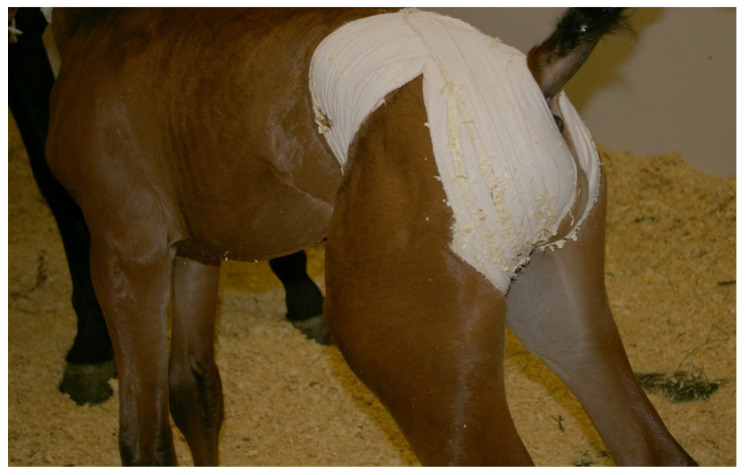
Congenital inguinal hernia in a thoroughbred foal managed by manual reduction and placement of truss.

**Figure 9 vetsci-11-00243-f009:**
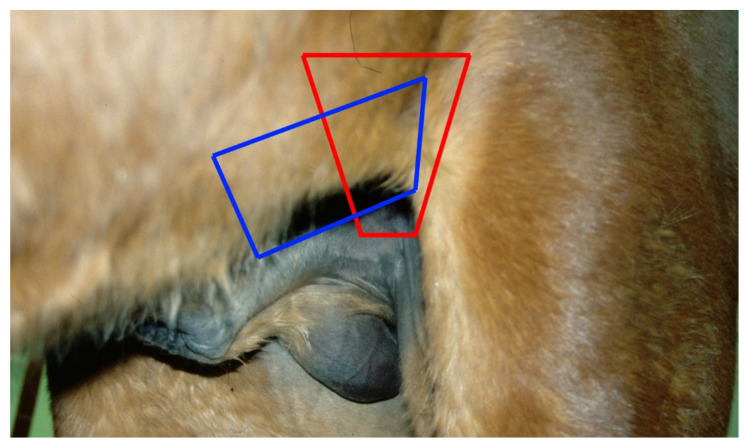
Transcutaneous ultrasound imaging zones to determine the location of a cryptorchid testis: inguinal imaging (red), caudo-cranial transabdominal imaging (blue).

**Figure 10 vetsci-11-00243-f010:**
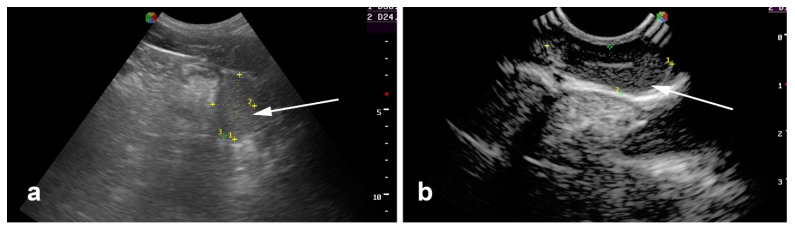
Transcutaneous inguinal ultrasonography. (**a**) Partial abdominal, and (**b**) complete abdominal cryptorchid testes.

**Figure 11 vetsci-11-00243-f011:**
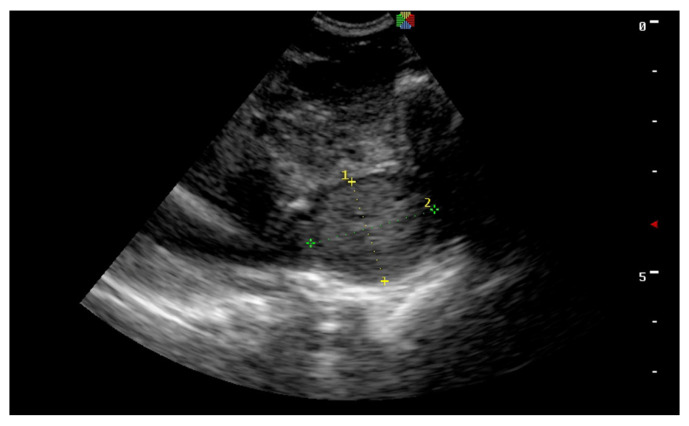
Transcutaneous transabdominal ultrasonography of a complete abdominal testis.

**Figure 12 vetsci-11-00243-f012:**
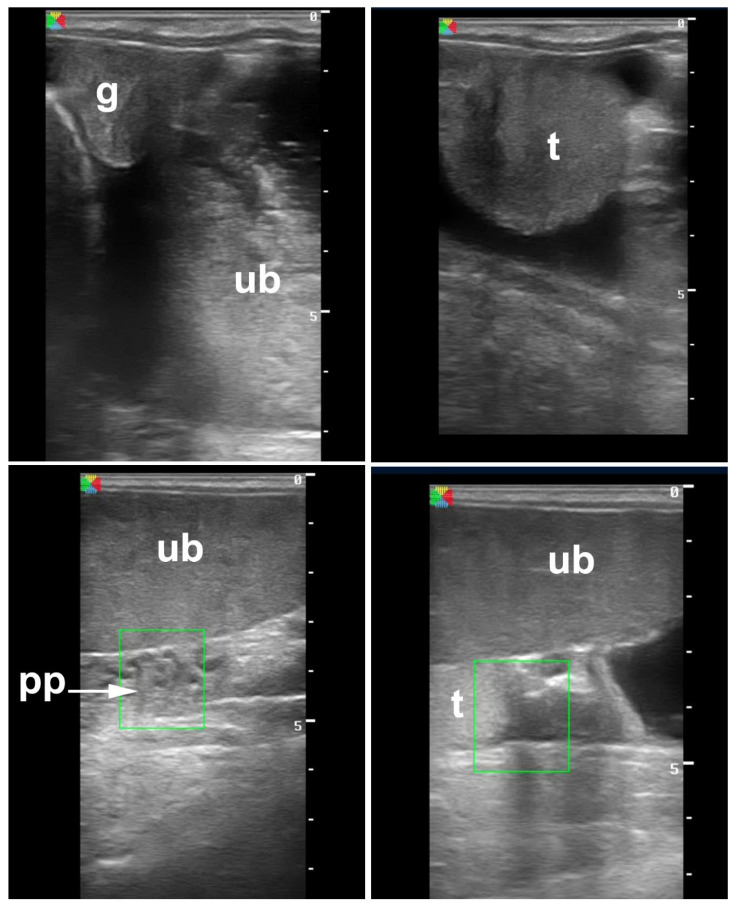
Transrectal ultrasonography of abdominal testis. g: Gubernaculum, ub: urinary bladder, t: testis, pp: pampiniform plexus.

**Figure 13 vetsci-11-00243-f013:**
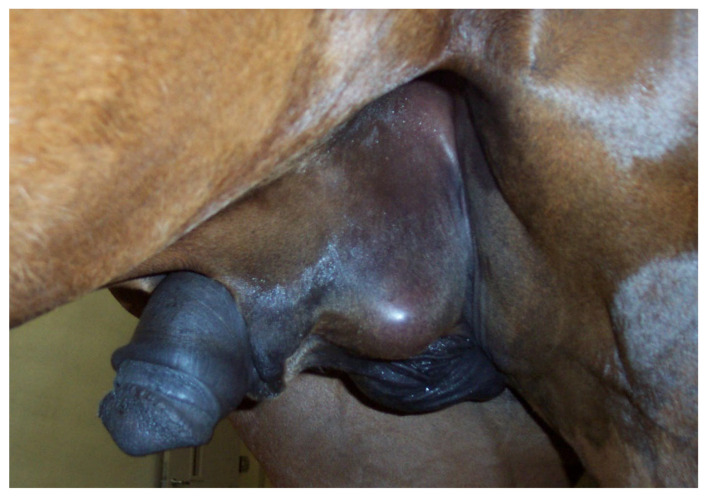
Appearance of the scrotum of stallion that sustained a testicular trauma (a kick) to the left testis during breeding.

**Figure 14 vetsci-11-00243-f014:**
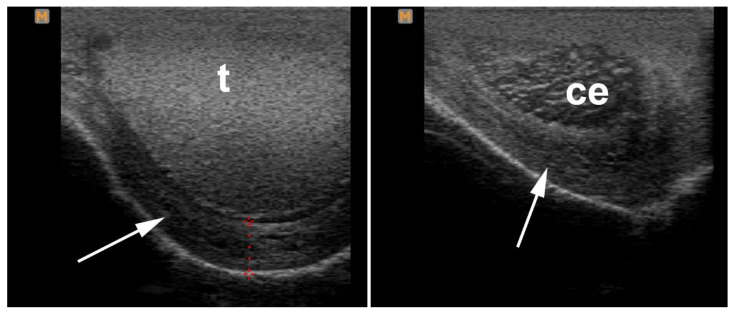
Ultrasonogram showing scrotal edema (arrow) following a traumatic injury. t: testis, ce: cauda (tail of) epididymis.

**Figure 15 vetsci-11-00243-f015:**
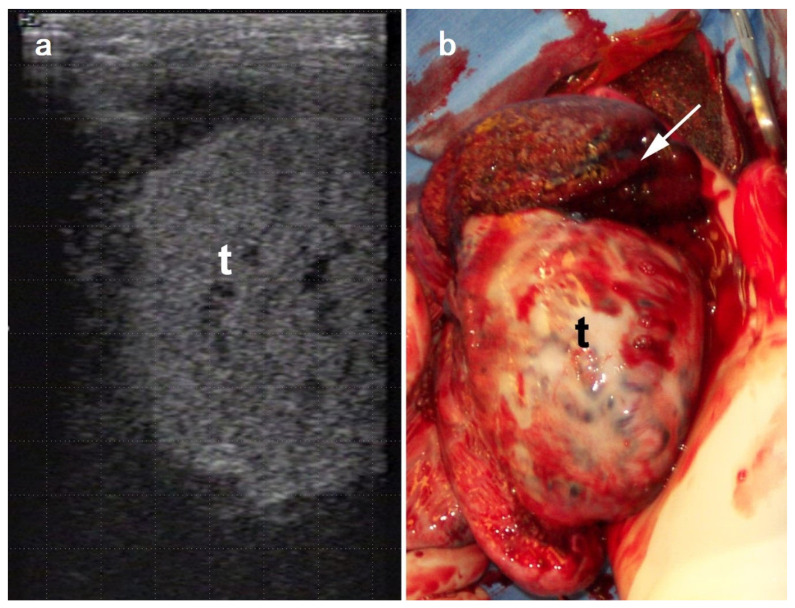
Testicular rupture and hemorrhage. (**a**) ultrasonographic image showing the irregular contour of the testis due to rupture albuginea, (**b**) ruptured testis (arrow) during the unilateral castration. (t: testis).

**Figure 16 vetsci-11-00243-f016:**
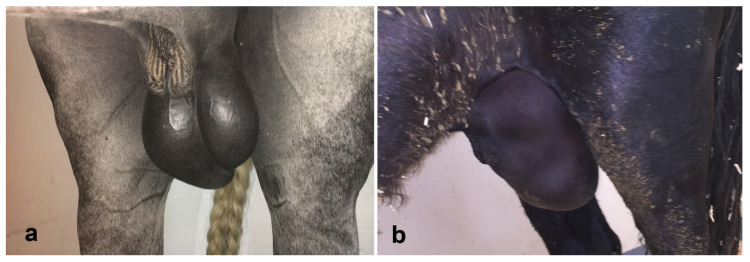
External appearance of the scrotum in 2 stallions. (**a**) Indirect Inguinal hernia of the jejunum in an Andalusian stallion, (**b**) inguinal hernia of the large intestine in a quarter horse stallion.

**Figure 17 vetsci-11-00243-f017:**
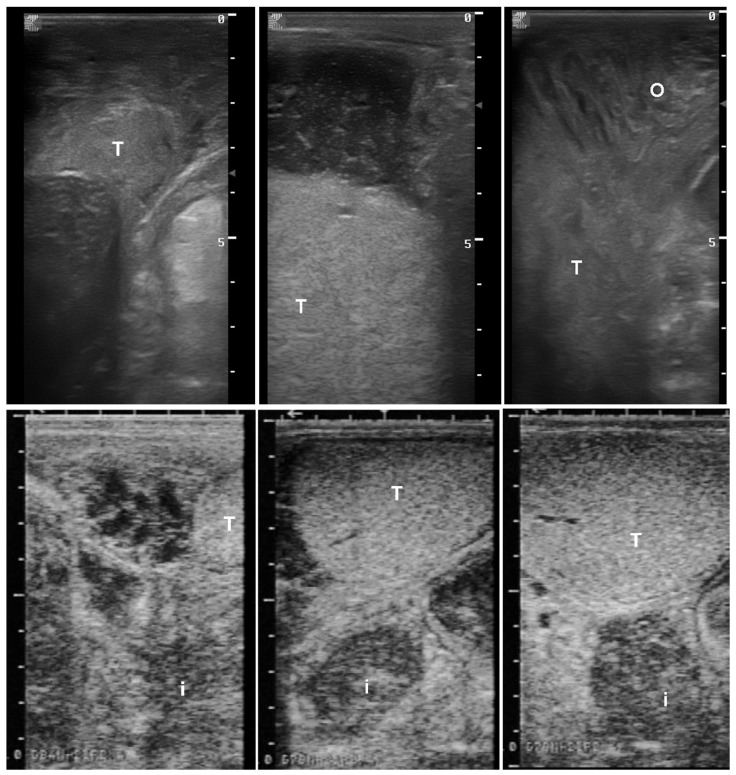
Ultrasonographic images of inguinal hernia of the omentum (**top**) and jejunum (**bottom**). T: testis, o: omentum, i: intestine).

**Figure 18 vetsci-11-00243-f018:**
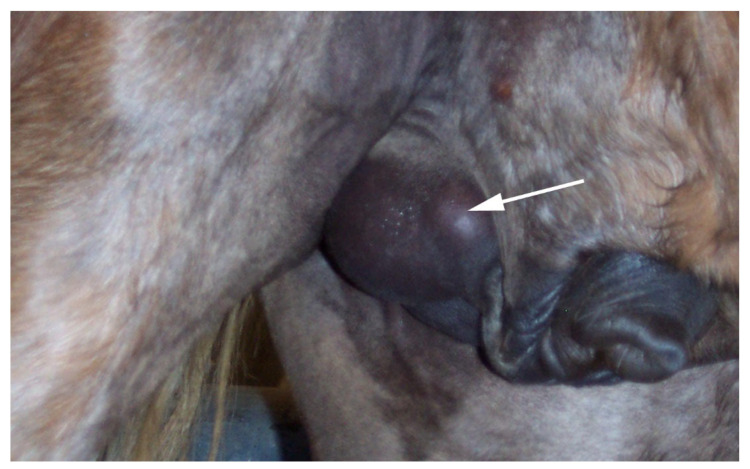
Right 180-degree spermatic cord torsion in a stallion. Note the cranial location of the tail of the epididymis (arrow).

**Figure 19 vetsci-11-00243-f019:**
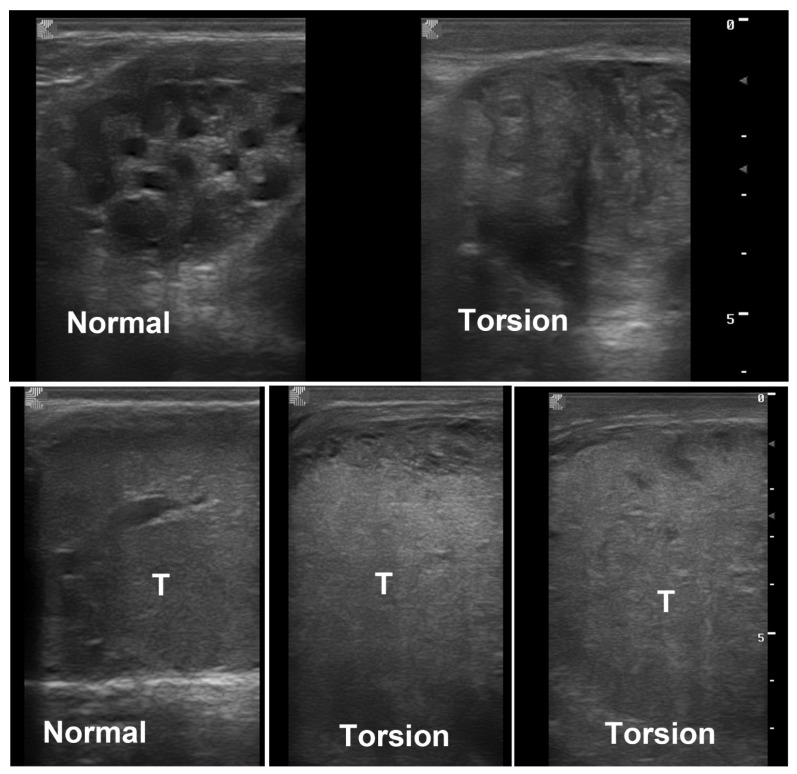
Ultrasonography of the spermatic cords (**top**) and the testes (**bottom**) in a stallion with a unilateral spermatic cord torsion. T: testis.

**Figure 20 vetsci-11-00243-f020:**
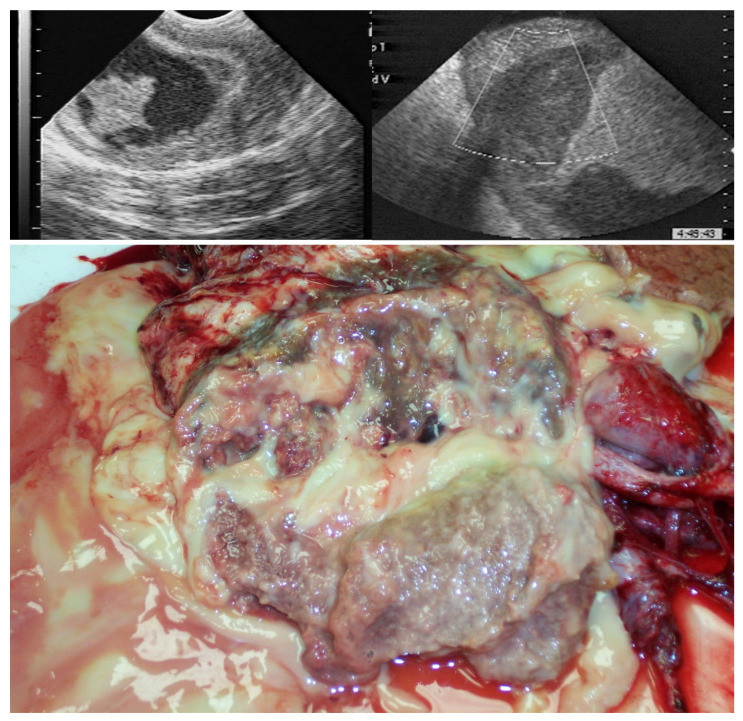
Ultrasonographic (**top**) and gross (**bottom**) of orchitis with abscessation of the testis due to Corynebacterium pseudotuberculosis in a stallion.

**Figure 21 vetsci-11-00243-f021:**
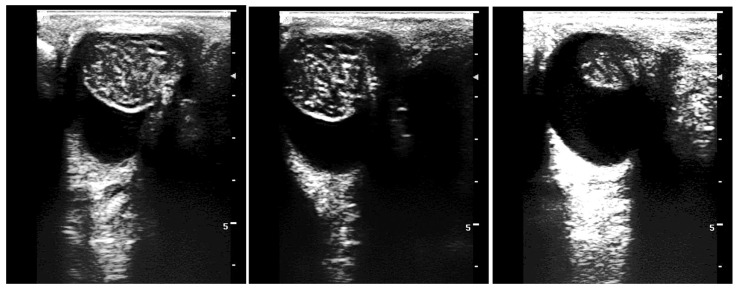
Ultrasonographic appearance of epididymitis in a stallion.

**Figure 22 vetsci-11-00243-f022:**
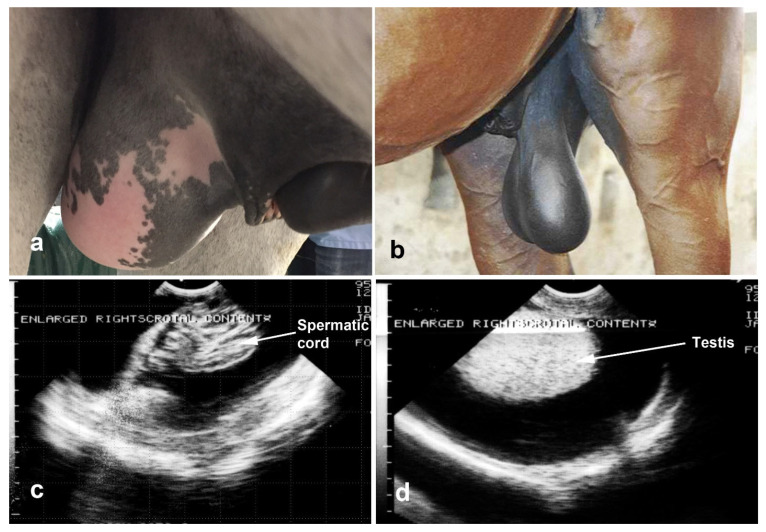
Gross (top) and ultrasonographic appearance of hydrocele in stallions in an environment of high heat index. (**a**) Moderate, (**b**) severe, (**c**) spermatic cord, (**d**) testis.

**Figure 23 vetsci-11-00243-f023:**
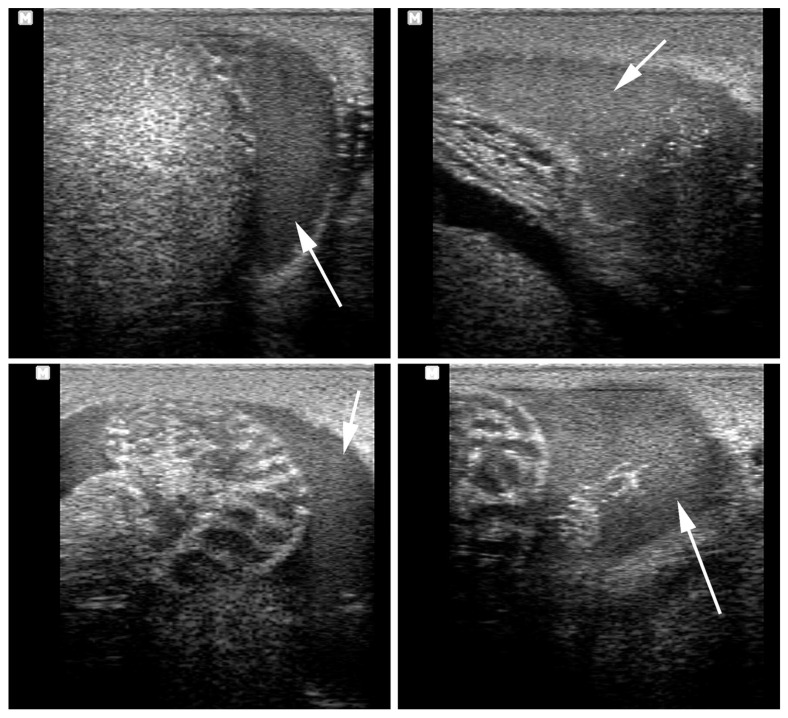
Ultrasonographic appearance of a pyocele (arrows) in a stallion with post-surgical peritonitis.

**Figure 24 vetsci-11-00243-f024:**
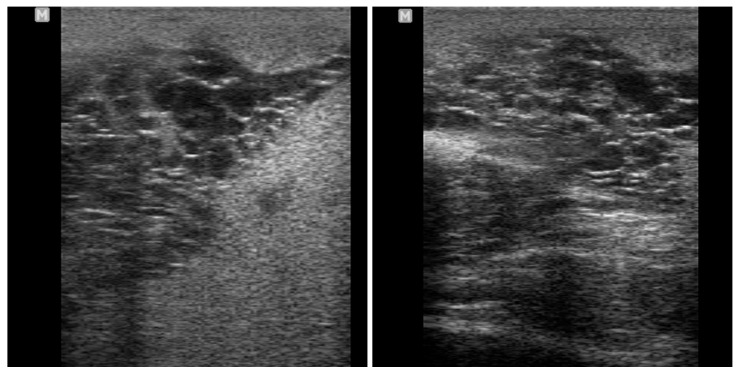
Ultrasonographic appearance of varicocele in a stallion.

**Figure 25 vetsci-11-00243-f025:**
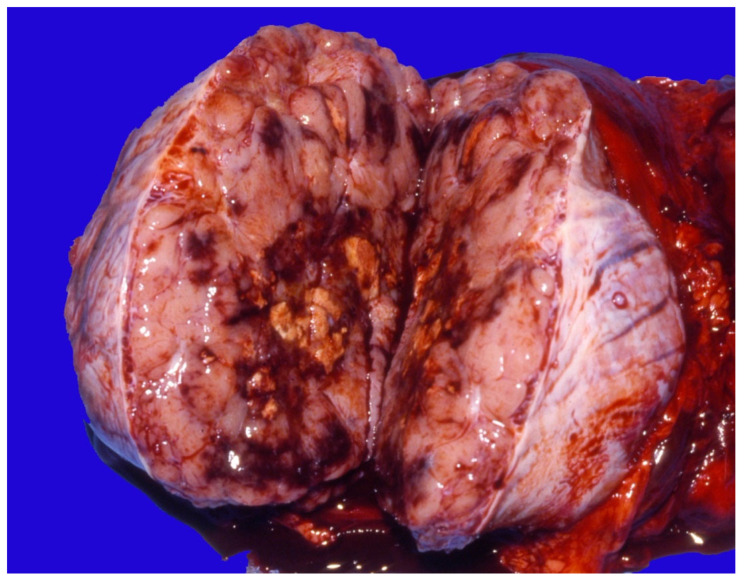
Gross appearance of testicular seminoma in a stallion.

**Figure 26 vetsci-11-00243-f026:**
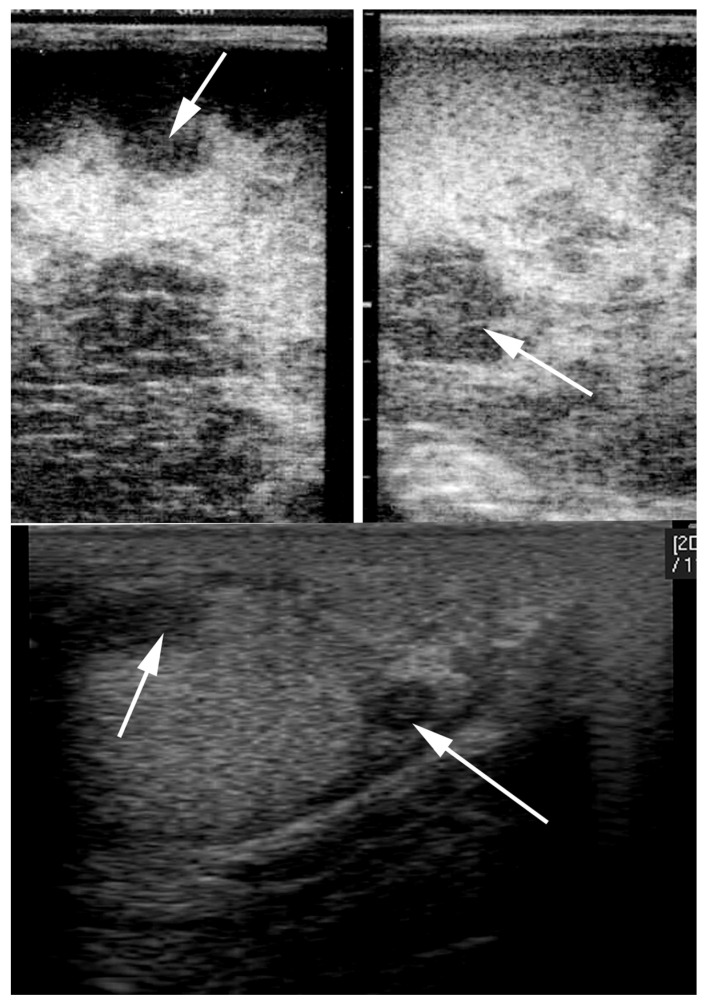
Ultrasonographic appearance of seminoma in a stallion.

**Figure 27 vetsci-11-00243-f027:**
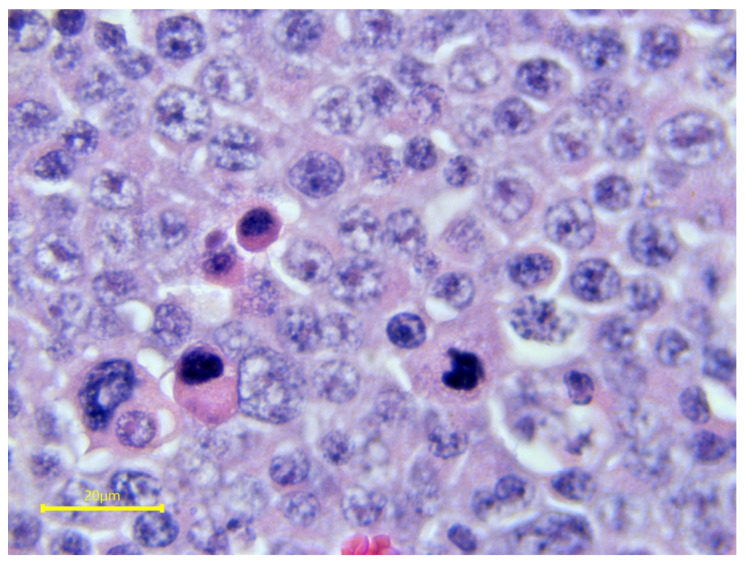
Histological features of testicular seminoma from a stallion.

**Figure 28 vetsci-11-00243-f028:**
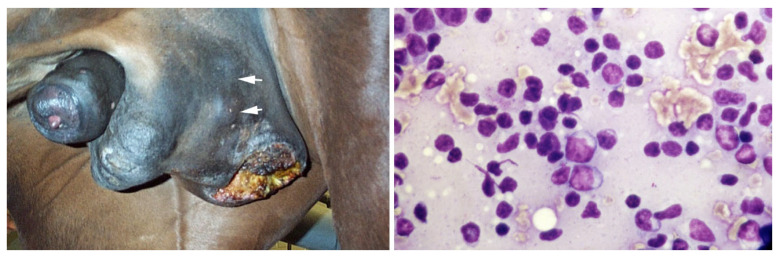
Gross appearance and fine needle aspirate from cutaneous scrotal lymphosarcoma (nodules indicated by arrows) in a Quarter horse stallion.

**Figure 29 vetsci-11-00243-f029:**
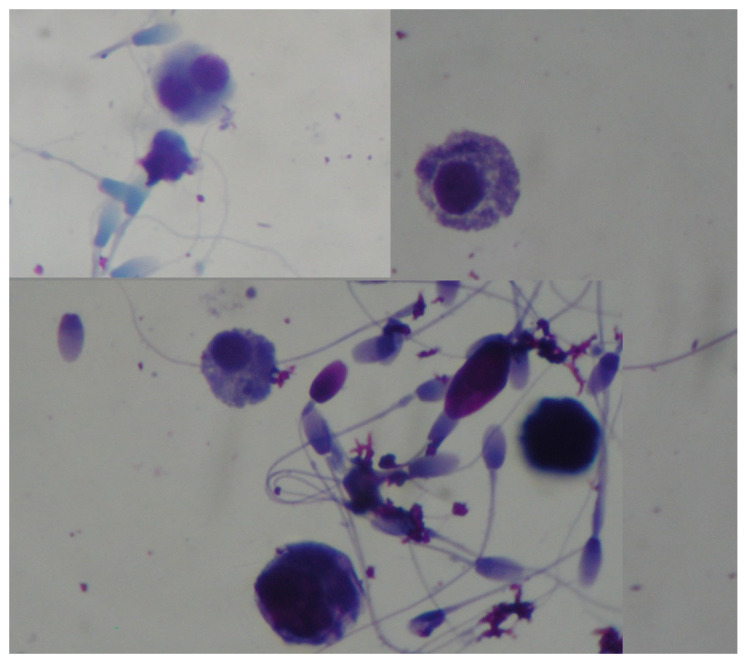
Premature round germ cells seen in ejaculate from a stallion with testicular degeneration.

**Figure 30 vetsci-11-00243-f030:**
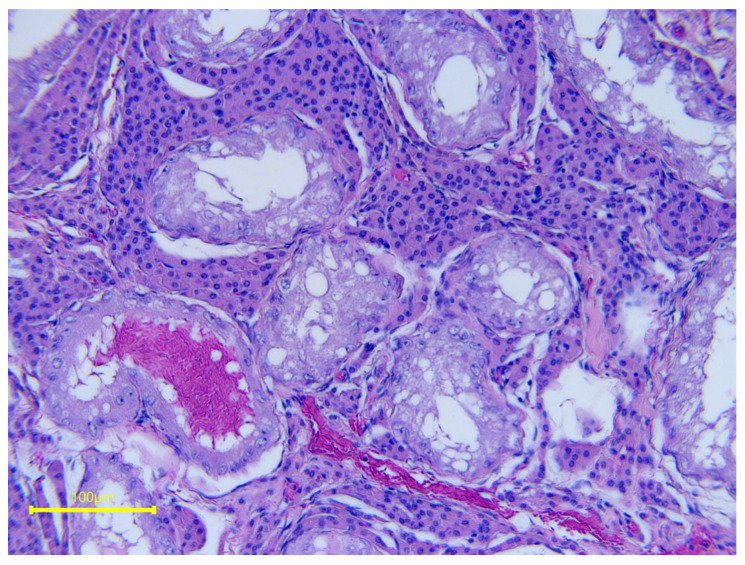
Histological features of severe testicular degeneration in a stallion.

**Table 1 vetsci-11-00243-t001:** Testicular volume and daily sperm output calculation.

Testicular Volume (cm^3^)	=4/3 ***π*** × (Length of Testis (cm)/2 × Width of Testis (cm)/2 × Height of Testis (cm)/2)
or Testicular volume (cm^3^)	=0.5233 × length × width × height
DSO (billions of sperm/day)	=(Total Testicular Volume) × 0.024 − 0.76

**Table 2 vetsci-11-00243-t002:** Mean percentages of germ cells and number of Sertoli cells/100 germ cells (±SD) observed in smears of testicular fine-needle aspiration from 15 horses [24].

Cell Type	Mean ± SD (%)
Spermatogonia	1.6 ± 1.1
Spermatocytes I	3.4 ± 2.2
Spermatocytes II	0.8 ± 0.7
Early spermatids	25.5 ± 9.5
Late spermatids	37 ± 9.3
Spermatozoa	31.5 ± 8.5
Sertoli cells/100 g cells	20.9 ± 17

**Table 3 vetsci-11-00243-t003:** Endocrine testing for cryptorchidism in horses based on data from the Clinical Endocrinology Laboratory, University of California Davis, Davis, CA, USA.

Hormone Tested	Stallions	Cryptorchids	Geldings	Sensitivity	Specificity
Testosterone (pg/mL)	>800	100–500	<50	85%	91%
Estrone sulfate (ng/mL)	140–200	35–60	<0.1	88%	84%
AMH (ng/mL)	30–200	>0.15	-	-	-

## Data Availability

Not applicable.

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
