# Peer review of "Diagnostic Approach to Equine Testicular Disorders"

_vetsci, 2024, doi:10.3390/vetsci11060243_

Round 1
Reviewer 1 Report
Comments and Suggestions for Authors
In my opinion the authors well describe the diagnostic process of testicular disease in the horse.
Author Response
Thank you for your time and effort to review the manuscript. We appreciate your help.
Reviewer 2 Report
Comments and Suggestions for Authors
line 142 and 373: please use always the same word, tail or cauda
line 239 testicular hypoplasia can also be acquired, please add
lines 246-247 "clinically the testes may be softer or harder than normal" any reference is provided, and if it's referred to the authors experience this statement is questionable. Please list the clinical condition related to both, harder or softer
Author Response
line 142 and 373: please use always the same word, tail or cauda
Thank you for highlighting this. “tail of” has been added to the line 373 considering that both terms are correct and have been used in the literature. The reader will benefit by learning both naming systems.
line 239 testicular hypoplasia can also be acquired, please add
Thank you for suggesting this. We would please like to keep it this way. In our opinion, the term hypoplasia is by definition the incomplete development or underdevelopment of an organ therefore it is congenital. The acquired condition is atrophy. We have discussed that under testicular degeneration.
lines 246-247 "clinically the testes may be softer or harder than normal" any reference is provided, and if it's referred to the authors experience this statement is questionable. Please list the clinical condition related to both, harder or softer
Thank you for pointing out. We appreciate that. It is our clinical experience, and the sentence has been edited/updated. NOW line 245-246.
Reviewer 3 Report
Comments and Suggestions for Authors
The Authors provide a complete review on the diagnostic approach to testicular disorders in stallions.
Here are some suggestions that the Authors could consider improving the manuscript.
Line 60: please change “2. . Examination techniques of the stallion scrotum and its content” with “2. Examination techniques of the stallion scrotum and its content”
Lines 104-105: please put the legend after the figure 2.
Line 133: please check the yellow evidenced part.
Line 173: the figure 5 is missing.
Lines 195-197: please put the legend after the figure 6.
Lines 214-218: an English revision is needed.
Lines 252: please check the format of citation.
Line 254: please check the figure 8, legend position and the format of the legend.
Line 255: please put the sentence before the figure 8.
Line 267: please check the citation.
Line 278: Due to the genetic component, the cryptorchidism is one of the condition for exclusion of stallion for reproduction. The Stud Book rules of most of the breeds, exclude cryptorchid stallions as selected reproducers.
Line 287: please check the citation.
Lines 300-305: please check the number of the table in the sentence and in the legend of the table.
Lines 333-334: please check the position of the legend.
Line 366: please check the citation.
Lines 371-374: please check the format of the two legends.
Please check the format of the legend for figures 15 to 31.
Line 496: please change “….. A Ischemia and oxidative injury……” with “…..An ischemia and an oxidative ……”.
Line 675: please change “…in testicular size [134].Acute colic and ….” with “…..in testicular size [134]. Acute colic and…..”.
Line 780: the figure 31 is missing.
Author Response
Line 60: please change “2. . Examination techniques of the stallion scrotum and its content” with “2. Examination techniques of the stallion scrotum and its content”
Changed. Thank you for highlighting
Lines 104-105: please put the legend after the figure 2.
Corrected
Line 133: please check the yellow evidenced part.
Thank you for highlighting. The reference format is corrected.
Line 173: the figure 5 is missing.
Thank you for highlighting. “figure 5 has been deleted” and all figure numbers have been cortrected.
Lines 195-197: please put the legend after the figure 6.
Corrected.
Lines 214-218: an English revision is needed.
Thank you for pointing out. The changes/corrections have been made in line 215 and 216.
Lines 252: please check the format of citation.
Thank you. Corrected in line 252
Line 254: please check the figure 8, legend position and the format of the legend.
Thank you. Corrections made in line 253 (NOW line 256)
Line 255: please put the sentence before the figure 8.
The suggested change is made and highlighted. Line 253-255
Line 267: please check the citation.
Thank you. The incorrect reference format is deleted. Now line 266.
Line 278: Due to the genetic component, the cryptorchidism is one of the condition for exclusion of stallion for reproduction. The Stud Book rules of most of the breeds, exclude cryptorchid stallions as selected reproducers.
Thank you for the suggestion. We greatly appreciate that. The information has been added (NOW line 278-279).
Line 287: please check the citation.
Thank you. Corrected. Now line 288.
Lines 300-305: please check the number of the table in the sentence and in the legend of the table.
Thank you for pointing out. Corrected in line 303 and 305.
Lines 333-334: please check the position of the legend.
Thank you. The legend is at correct place. It may have been due to downloading in MS Word format in different systems. The figure ends at line 333 and the legend is in line 334-335.
Line 366: please check the citation.
The style corrected. Now line 367.
Lines 371-374: please check the format of the two legends.
Font size for both figures corrected. Line 371-372, and line 373-374.
Please check the format of the legend for figures 15 to 31.
All figure legends are corrected (font corrected), and changes highlighted.
Line 496: please change “….. A Ischemia and oxidative injury……” with “…..An ischemia and an oxidative ……”.
Thank you for pointing out. Corrected. Now line 494
Line 675: please change “…in testicular size [134].Acute colic and ….” with “…..in testicular size [134]. Acute colic and…..”.
Thank you. It is corrected, NOW line 673
Line 780: the figure 31 is missing.
Thank you for pointing out. The figure has been added (Now line 778-779)
Reviewer 4 Report
Comments and Suggestions for Authors
This is an excellent overview of conditions affecting the testes and scrotum, and their diagnostics. It is very well-written and engaging to read. I don’t have any major suggestions, and only offer the following small tweaks:
· Line 71: isn’t sudden enlargement a clinical sign? Consider rewording to say “sudden enlargement without other clinical signs”
· Line 133: Please update the citation.
· Line 267: Please update the citation.
· Line 287: Please update the citation.
· Line 366: Please update the citation.
· Line 444: Please update the citation.
· Line 625: A space is needed after tumors: “tumors( Leydig cell…”
· Table 1: There seem to be too many right-seded parentheses in the formula for testicular volume. Please confirm the equation.
· Figure 15: This figure could be enhanced if there were also a comparison to a normal scrotum. For example, there could be an image from a normal scrotum as part of this figure, or the normal range for scrotal wall thickness could be listed in the text (on line 369), and then the measurement illustrated in red on the left image could be presented to illustrate that the wall is thickened.
Author Response
This is an excellent overview of conditions affecting the testes and scrotum, and their diagnostics. It is very well-written and engaging to read. I don’t have any major suggestions, and only offer the following small tweaks:
Thank you for your time and effort to review.
- Line 71: isn’t sudden enlargement a clinical sign? Consider rewording to say “sudden enlargement without other clinical signs”
Thank you. “other is added to the line 71.
- Line 133: Please update the citation.
updated
- Line 267: Please update the citation.
updated
- Line 287: Please update the citation.
updated
- Line 366: Please update the citation.
updated
- Line 444: Please update the citation.
updated
- Line 625: A space is needed after tumors: “tumors( Leydig cell…”
Thank you. It is added. NOW line 623
- Table 1: There seem to be too many right-seded parentheses in the formula for testicular volume. Please confirm the equation.
Thank you for pointing out. The parenthesis are corrected now. NOW line 106
- Figure 15: This figure could be enhanced if there were also a comparison to a normal scrotum. For example, there could be an image from a normal scrotum as part of this figure, or the normal range for scrotal wall thickness could be listed in the text (on line 369), and then the measurement illustrated in red on the left image could be presented to illustrate that the wall is thickened.
Thank you for suggesting this. We considered this and decided that because figures 3 and 4 represent normal so adding another normal to figure 15 (now figure 14) will be a repetition. We hope that this can work please.
Reviewer 5 Report
Comments and Suggestions for Authors
Overall, the work was well organized and comprehensively described;The information given in the text provides a good background as well as perspectives on Diagnostic approach to equine testicular disorders
Comments and Suggestions for Authors
It´s recommended to use arrow or letter codes to clearly indicate a structure or region in the following images:
Figure 22: Ultrasonographic appearance of epididymitis in a stallion
Figure 25: Ultrasonographic appearance of varicocele in a stallion
Figure 28: Histological features of testicular seminoma from a stallion In the absence of metastasis, hemicastration is the treatment of choice in cases of testicular tumors.
Figure 29: Histological picture of cutaneous scrotal lympho sarcoma
Figure 30: Premature round germ cells seen in ejaculate from a stallion with testicular degeneration.
Author Response
Overall, the work was well organized and comprehensively described;The information given in the text provides a good background as well as perspectives on Diagnostic approach to equine testicular disorders
Thank you for your time and effort to review.
Comments and Suggestions for Authors
It´s recommended to use arrow or letter codes to clearly indicate a structure or region in the following images:
Thank you for suggesting. We will like to highlight that wherever multiple organs or confuisn is possible, we have added arrows or letters to figures, e.g., NOW figures 3,4,7,9,10,11,12,14, 15, 17,18,19,22,23,26> we have not added to the figures where it seemed unnecessary.
Figure 22: Ultrasonographic appearance of epididymitis in a stallion
Thank you for suggesting. We will like to highlight that wherever multiple organs or confuisn is possible, we have added arrows or letters to figures, e.g., NOW figures 3,4,7,9,10,11,12,14, 15, 17,18,19,22,23,26> we have not added to the figures where it seemed unnecessary.
Figure 25: Ultrasonographic appearance of varicocele in a stallion
Thank you for suggesting. We will like to highlight that wherever multiple organs or confuisn is possible, we have added arrows or letters to figures, e.g., NOW figures 3,4,7,9,10,11,12,14, 15, 17,18,19,22,23,26> we have not added to the figures where it seemed unnecessary.
Figure 28: Histological features of testicular seminoma from a stallion In the absence of metastasis, hemicastration is the treatment of choice in cases of testicular tumors.
Thank you for suggesting. We will like to highlight that wherever multiple organs or confuisn is possible, we have added arrows or letters to figures, e.g., NOW figures 3,4,7,9,10,11,12,14, 15, 17,18,19,22,23,26> we have not added to the figures where it seemed unnecessary.
Figure 29: Histological picture of cutaneous scrotal lympho sarcoma
Thank you for suggesting. We will like to highlight that wherever multiple organs or confuisn is possible, we have added arrows or letters to figures, e.g., NOW figures 3,4,7,9,10,11,12,14, 15, 17,18,19,22,23,26> we have not added to the figures where it seemed unnecessary.
Figure 30: Premature round germ cells seen in ejaculate from a stallion with testicular degeneration.
Thank you for suggesting. We will like to highlight that wherever multiple organs or confuisn is possible, we have added arrows or letters to figures, e.g., NOW figures 3,4,7,9,10,11,12,14, 15, 17,18,19,22,23,26> we have not added to the figures where it seemed unnecessary.